# Human Osteoblasts’ Response to Biomaterials for Subchondral Bone Regeneration in Standard and Aggressive Environments

**DOI:** 10.3390/ijms241914764

**Published:** 2023-09-29

**Authors:** Stefania Pagani, Manuela Salerno, Giuseppe Filardo, Janis Locs, Gerjo J.V.M. van Osch, Jana Vecstaudza, Laura Dolcini, Veronica Borsari, Milena Fini, Gianluca Giavaresi, Marta Columbaro

**Affiliations:** 1Surgical Sciences and Technologies, IRCCS Istituto Ortopedico Rizzoli, 40136 Bologna, Italy; stefania.pagani@ior.it (S.P.); veronica.borsari@ior.it (V.B.); gianluca.giavaresi@ior.it (G.G.); 2Applied and Translational Research Center, IRCCS Istituto Ortopedico Rizzoli, 40136 Bologna, Italy; 3Rudolfs Cimdins Riga Biomaterials Innovations and Development Centre of RTU, Institute of General Chemical Engineering, Faculty of Materials Science and Applied Chemistry, Riga Technical University, LV-1007 Riga, Latvia; janis.locs@rtu.lv (J.L.); jana.vecstaudza@rtu.lv (J.V.); 4Department of Orthopedics and Sports Medicine, Erasmus MC, University Medical Center Rotterdam, 3015 GD Rotterdam, The Netherlands; g.vanosch@erasmusmc.nl; 5Department of Otorhinolaryngology, Head and Neck Surgery, Erasmus MC, University Medical Center Rotterdam, 3015 GD Rotterdam, The Netherlands; 6Department of Biomechanical Engineering, Delft University of Technology, 2628 CD Delft, The Netherlands; 7Fin-Ceramica Faenza S.p.A, 48018 Faenza, Italy; ldolcini@finceramica.it; 8Scientific Direction, IRCCS Istituto Ortopedico Rizzoli, 40136 Bologna, Italy; milena.fini@ior.it; 9Electron Microscopy Platform, IRCCS Istituto Ortopedico Rizzoli, 40136 Bologna, Italy; marta.columbaro@ior.it

**Keywords:** osteochondral regeneration, human osteoblasts, biomimetic scaffold, oxidative stress

## Abstract

Osteochondral lesions, when not properly treated, may evolve into osteoarthritis (OA), especially in the elderly population, where altered joint function and quality are usual. To date, a collagen/collagen–magnesium–hydroxyapatite (Col/Col-Mg-HAp) scaffold (OC) has demonstrated good clinical results, although suboptimal subchondral bone regeneration still limits its efficacy. This study was aimed at evaluating the in vitro osteogenic potential of this scaffold, functionalized with two different strategies: the addition of Bone Morphogenetic Protein-2 (BMP-2) and the incorporation of strontium (Sr)-ion-enriched amorphous calcium phosphate (Sr-ACP) granules. Human osteoblasts were seeded on the functionalized scaffolds (OC+BMP-2 and OC+Sr-ACP, compared to OC) under stress conditions reproduced with the addition of H_2_O_2_ to the culture system, as well as in normal conditions, and evaluated in terms of morphology, metabolic activity, gene expression, and matrix synthesis. The OC+BMP-2 scaffold supported a better osteoblast morphology and stimulated scaffold colonization, cell activity, and extracellular matrix secretion, especially in the stressed culture environment but also in normal culture conditions, with increased expression of genes related to osteoblast differentiation. In conclusion, the incorporation of BMP-2 into the Col/Col-Mg-HAp scaffold also represents an improvement of the osteochondral scaffold in more challenging conditions, supporting further preclinical studies to optimize it for use in clinical practice.

## 1. Introduction

The osteochondral unit is pivotal in joint lubrication and the transmission of joint constraints to bones during movement [1]. Osteochondral lesions derived from traumatic injuries or degenerative lesions, which both increase due to higher physical activity in the aging population, represent a debilitating and painful condition [2]. If not properly treated, osteochondral lesions can evolve into osteoarthritis (OA), in the worst cases requiring drastic joint replacement surgeries with huge impacts on patient health and the economy [3,4]. Any treatment that addresses the repair of osteochondral lesions should reflect the anatomical and functional features of the whole osteochondral unit, which is composed of two different tissues: the articular cartilage and the underlying subchondral bone. Due to the complex multiphasic composition and structure of the osteochondral unit, the successful regeneration of these lesions is particularly difficult [5,6]. This is even more challenging in the elderly population [7], where age-related loss of normal joint homeostasis implies altered joint function and quality, with changes in bone structure and biomechanical properties as well as in the overall articular environment [8].

Regenerative scaffold-based procedures have emerged as a potential option to address osteochondral lesions [9], and a bilayered collagen–hydroxyapatite (HA) scaffold mimicking native osteochondral compositions and structures has already proven to be a biocompatible and clinically effective strategy [10,11]. The scaffold is composed of a superficial cartilage-like layer (collagen only) and a deep bone-like layer (collagen mixed with Mg-HA), addressing the repair of the underlying subchondral bone. However, despite very good clinical results in terms of cartilage formation, suboptimal subchondral bone regeneration has been documented as a limit of this scaffold [12]. Given the increasing number of elderly OA patients with osteochondral lesions [7], investigating new approaches to support subchondral bone regeneration in a more compromised environment is paramount, with particular regard to osteoblast function, whose dysfunction has a crucial role in the pathogenesis of subchondral bone degeneration [13,14]. To overcome the suboptimal bone-healing potential of this clinically used OC scaffold, two possible strategies for its functionalization could be promising.

On one side, the addition of the well-known osteogenic factor Bone Morphogenetic Protein-2 (BMP-2), which has been approved by the FDA for human clinical applications [15], has already been applied in several in vitro and in vivo models, with promising results [16,17,18]. On the other side, the incorporation of nanostructured amorphous calcium phosphate (ACP) granules with strontium (Sr) ions combines the known properties of biocompatibility, bioactivity, and osteoconductivity of ACP and those of the Sr ion, obtaining a potential therapeutic agent for bone repair [19,20,21]. The regenerative effects of these strategies could be limited, or one strategy could be better than the other one in compromised situations such as osteoarthritis or aging. Aging-related conditions, as well as OA, have been associated with an oxidative stress status derived from the accumulation of reactive oxygen species (ROS), among which is hydrogen peroxide [22]. Therefore, understanding the responses of osteoblasts grown on functionalized scaffolds under stressful conditions would help the understanding of their potential application in patients of all ages. The method chosen to reproduce this condition in vitro consisted of the addition of H_2_O_2_ to the culture system, a previously established model in several in vitro studies [23,24,25,26].

Thus, the aim of this study was to evaluate the behavior of human osteoblasts grown on the bone-like layer of the OC scaffold modified by two different strategies, the addition of BMP-2 and the incorporation of Sr-ACP granules, in normal and stressful in vitro conditions.

## 2. Results

### 2.1. Effects of the Modified Scaffolds on Osteoblast Morphology

To explore the osteoinductive properties of the modified scaffolds, a comparison of the materials in normal conditions was performed. The cells grown on the scaffolds were analyzed in terms of morphology, cell viability, and gene expression. The different cell/scaffold constructs were compared after H&E staining, showing higher cell numbers on the OC and OC+BMP-2 scaffolds compared to the OC+Sr-ACP scaffold. With the first one, a gradual migration toward the inner part of the scaffold layer was observed (blue arrows) (Figure 1a), while with the OC+BMP-2, the cells were flattened, well-organized, and interconnected by a matrix (red arrows) (Figure 1a). An abundant matrix synthesis is fundamental in correct osteoblast physiology, but in a static seeding system, an external layer could be a partial limit, hampering a complete exchange of metabolites and cell entry within the biomaterial. A similar phenomenon but with milder characteristics was observed on the OC+Sr-ACP scaffold: a small cell number, connected by a thin matrix layer, on the material’s top (red arrows) (Figure 1a).

To further investigate the effects of the three different scaffolds on NHOst cells, we performed a TEM analysis. Our observations revealed good cell morphologies consisting of abundant rough endoplasmic reticula (rer), well-preserved mitochondria (m) and nuclei characterized by heterochromatin compartments (hc) close to the nuclear envelope, and euchromatin compartments (ec) scattered throughout the remaining part of the nucleus. We also noticed that the cells seeded on the OC+Sr-ACP biomaterial showed early features of alterations to the nucleus, with rarefied chromatin (r) and cytoplasmic vacuoles (v) ascribable to stress conditions. Interestingly, we also observed numerous focal contacts (black arrows) between the cell membrane and the hydroxyapatite integrated into the scaffold surfaces or Sr-ACP granule surfaces, demonstrating that the interaction of cells/biomaterials was encouraged. In other words, elements such as hydroxyapatite and Sr-ACP were revealed to be improvements in this aspect (Figure 1(bD–bF)).

### 2.2. Effects of the Modified Scaffolds on Osteoblast Morphology after Treatment with H_2_O_2_

To investigate the possible effect of the modified scaffold in a stress environment, H_2_O_2_ was added to the cultures. The effective H_2_O_2_ concentration and exposure time were set up based on preliminary experiments. In particular, the H_2_O_2_ caused decreases in cell metabolic activity and in ALPL, COL1A1, BGLAP, and SPARC, as well as increases in COX2 and iNOS (Appendix A). With similar results to what has been observed without treatment, both H&E staining and ultrastructural approaches were used to evaluate cellular behavior and distribution after exposure to the H_2_O_2_. A preliminary observation of the unstained scaffold sections revealed no appreciable differences in the compositions of the untreated and H_2_O_2_-treated OC and OC+BMP-2 scaffolds, although large granules were visible within the Sr-ACP biomaterial (Appendix A).

On the other side, the OC+Sr-ACP scaffold appeared to be different after the H_2_O_2_ treatment (Appendix A). H&E staining confirmed the heterogenicity and the presence of more void spaces in the OC+Sr-ACP scaffold in comparison to the OC and OC+BMP-2, while the cell density was rarefied in all materials compared to the untreated conditions (Figure 2a). In TEM, cells seeded on the OC+Sr-ACP scaffold showed cytoplasmic changes in terms of increases in vacuole numbers and mitochondrial swelling (Figure 2(bB)) that could be attributable to the necrosis process despite preserving the focal contacts between the cell membrane and hydroxyapatite integrated into the scaffold, as in the untreated sample (Figure 2(bE), black arrows). The cells seeded on the OC and OC+BMP-2 scaffold revealed an increase in autophagic vacuoles (av) (Figure 2(bA,bC)) attributable to stress conditions, probably due to the treatment with H_2_O_2_. The focal contacts between the cell membrane and scaffold were preserved (Figure 2(bD,bF), black arrows), as in the untreated scaffold.

### 2.3. Effects of the Modified Scaffolds on Osteoblast Metabolic Activity

The cell metabolic activity on the untreated scaffold, quantified with the Alamar Blue test, revealed significantly higher values with OC and OC+BMP-2 than with Sr-ACP (Figure 3). This result reflects what emerged in the histological staining (H&E) and in the TEM analysis of cell colonization (Figure 1). The metabolic activity of the untreated cells seeded on the OC+Sr-ACP scaffold was lower than that of the cells seeded on other scaffolds. After treatment with H_2_O_2_, the metabolic activity was lower than in the untreated samples, with the lowest activity in the OC+Sr-ACP constructs, while the NHOsts grown on the OC+BMP-2 also showed higher activity compared with those grown on the OC scaffold (Figure 3). In the OC+Sr-ACP construct, treatment with H_2_O_2_ resulted in a 70% reduction in cell activity. In the OC+BMP-2 construct, metabolic activity decreased by less than 40% following treatment with H_2_O_2_.

### 2.4. Effects of the Modified Scaffolds on Gene Expression

In normal culture conditions, the differentiation-related markers ALPL, COL1A1, and SPARC tended to be more expressed by the cells grown on the OC and OC+BMP-2 scaffolds than on the OC+Sr-ACP scaffold. Conversely, the expressions of COX2 and iNOS2, involved in the inflammation, were higher in the cells seeded on the OC+Sr-ACP scaffold, although these differences did not reach statistical significance between the samples (Figure 4).

After the H_2_O_2_ treatment, a significantly higher expression of ALPL was observed in the cells cultured on the OC+Sr-ACP scaffold, both vs. untreated cells grown on the same scaffold and vs. H_2_O_2_-treated cells grown on the OC scaffold. The H_2_O_2_-treated NHOsts grown on the OC+BMP-2 scaffold had a higher ALPL expression than that of the untreated cells (Figure 4). No significant differences were found for the osteogenic markers COL1A1, BGLAP, or SPARC, with a trend of higher BGLAP expression after treatment with H_2_O_2_ in all scaffolds. A general increase was observed in the inflammation-related markers COX2 and iNOS2 following treatment with H_2_O_2_ in all scaffolds.

### 2.5. Effects of the Modified Scaffolds on Protein Matrix Expression

To observe the matrix deposition, in particular its main component, Type I Collagen, both IHC staining and TEM analysis were performed. After the IHC staining, a thick and uniform lane was found on the seeding surface of the OC+BMP-2 scaffold. On the OC and OC+Sr-ACP biomaterials, a similar but thinner and irregular lane was observed, consistently with the H&E staining (Figure 5a). Ultrastructural analysis confirmed the presence of an abundant deposition of collagen fibrils (coll) randomly distributed and not oriented throughout the extracellular space in all the scaffolds. Additionally, small vesicles (black arrows) were evident among the collagen fibrils, which could be attributed to matrix vesicles deputed to mineralize the matrix through hydroxyapatite crystals (Figure 5b).

After exposure to H_2_O_2_, IHC staining showed the loss on all of the constructs of the cellular organization and of the line of the collagenic matrix on the seeding surface as well as inside the scaffold. On the contrary, numerous stained cells were visible, which is attributable to the collagen synthesis by the cells (Figure 6a). TEM analysis confirmed an increase in matrix deposition by the NHOsts seeded on the OC+BMP-2 scaffold compared with those of the OC scaffold after the H_2_O_2_ treatment. In addition, numerous small vesicles were observed among the collagen fibrils, especially in the OC+BMP-2 scaffold (Figure 6b, black arrows). This could be related to vesicles deputed to matrix mineralization through hydroxyapatite crystals. On the contrary, a lower collagen matrix was secreted by the NHOst cells seeded on the OC+Sr-ACP scaffold, and a scarce presence of small vesicles compared with those in the cells seeded on the other scaffolds was observed, probably due to H_2_O_2_-induced cell stress (Figure 6b, asterisks).

## 3. Discussion

The main results of this study suggest that adding BMP-2 to the bone-like layer of an OC scaffold improves osteoblast activity and the secretion of the autologous extracellular matrix, especially in an aggressive environment.

This work is focused on analysis of the bone regeneration potential of the bone-like layer of a clinically used scaffold for treating osteochondral lesions [10,11]. This scaffold layer, developed to improve subchondral bone regeneration, has a porous nano-structured composition aimed at the efficient delivery of ions, such as Sr, Ca, and HA, retained within the layer, favoring chemotaxis, scaffold colonization, and the cell mineralization process. Bio-active Mg ions have been introduced in the mineral phase to enhance the affinity of HA with natural bone [27] and promote an increase in osteogenic activity [28]. However, despite excellent clinical results being reached in terms of cartilage tissue repair, the poor regeneration of subchondral bone limited the success of this strategy [11,12], which is even more relevant in light of the growing number of elderly patients affected by osteochondral lesions with lower regenerative capacity [7,8]. It is important to consider that the cellular microenvironment in this patient population is characterized by inflammatory mediators and oxygen radicals, which keep cells in a state of continuous stress [29]. Additionally, OA and osteoporosis, two of the most common pathological conditions that affect bone tissue in the elderly population, show excessive accumulation of reactive oxygen species (ROS) [30]. In addition to these aspects related to inflammation, it is also necessary to consider observed imbalances within the bone marrow in favor of adipose tissue. Decreases in osteoblast numbers and in the differentiation ability of mesenchymal cells from the bone marrow niche (BMSCs) have been confirmed in several studies [31,32]. This can be partially explained by osteoblasts’ hyperactivity in stress environment [33] or by the good ability of adipose-derived stromal cells (ADSCs) to maintain this differentiating capacity [34]. In this light, two modifications, the addition of BMP-2 or granules containing amorphous calcium phosphate with a high specific surface area enriched with Sr ions (Sr-ACP), were applied to the bony layer of the scaffolds, and their effects on osteoblast behavior were evaluated by reproducing an aggressive in vitro environment by a short exposure to H_2_O_2_.

BMP-2 has long been known to be effective in stimulating the mineralization process of osteoblasts [35] and has been proposed as a strategy to enhance the osseointegration of several biomaterials [36]. Metallic alloys, polymers, and collagenic scaffolds, sometimes also coated with calcium phosphate as hydroxyapatite or in other formulations, are enhanced when enriched with BMP-2 on their surfaces, showing better bone growth in vivo [37]. There is also substantial agreement on the positive osteogenic role of BMP-2 in aged microenvironments [38,39,40]. In light of these remarks about alterations in the bone marrow microenvironment and the characteristics of mesenchymal cells in an aged context, the addition of BMP-2 to a Col/Col-Mg-HAp scaffold can be an advantage. Other BMPs are known to also possess strong osteogenic potential. Among these, for example, BMP-9 has recently emerged as the strongest one [41]. Unfortunately, a lack of heparin-binding motifs to enable interactions with ECM [42] makes it unsuitable for this purpose. BMP-2 was therefore chosen for this study, thanks also to its current use in clinical practice [15]. Similarly, Sr-amorphous calcium phosphate has been proposed as a bioinspired additive (or substitute) that combines major bone mineral features with the dual effects exerted by Sr: bone resorption reduction and increase in bone formation [43]. Together with a biomimetic scaffold that mimics bone structure thanks to its composition (Type I Collagen and hydroxyapatite), the proposed strategies have been conceived and applied to improve the limited subchondral bone regeneration potential of scaffolds.

Although the clinical outcomes of the patients treated with the unmodified scaffold were not influenced by age [11,44], the two strategies to improve the bone-like layer of the scaffold might help in creating a favorable milieu for osteoblast functionality in an aged condition. To observe the effects of these biomaterials, with a particular focus on osteogenesis, human primary osteoblasts were used. This should represent the human cells responsible for bone deposition within the osteochondral unit and involved in the dysfunction of OA subchondral bone [45]. Several authors have used these cells to test innovative biomaterials aimed at substituting bone tissue [46,47]. The first part of this study, which compared the three scaffolds under normal culture conditions, provides a preliminary indication of cell viability and the expressions of typical osteoblast genes as well as of the inflammatory genes. This revealed similar behavior in cells grown on the OC and OC+BMP-2 scaffolds. The expressions of genes that regulate osteoblast activity, which consists of the synthesis and deposition of the intercellular matrix, did not have any highlighted significant differences between the two groups. However, an organized layer of the intercellular matrix on the top of the OC+BMP-2 scaffold was more evident than on the OC scaffold. Conversely, as observed with IHC, the Type I Collagen was less organized and abundant on OC-Sr-ACP. With TEM, collagen fibrils were found to be similar and randomly arranged on all the biomaterials, including OC+Sr-ACP. In light of the cell morphological features and molecular biology data, the addition of the BMP-2 to the OC scaffold seemed to improve the basal biomaterial potential, although it did not promote the activation of osteoblast-activity-related genes. A possible reason could be that NHOsts are mature osteoblasts and, unlike their progenitors mesenchymal stromal cells, are less stimulated to express these genes.

The functionalized scaffolds were then tested in a challenging condition that reflected a more stressful cell environment. For this purpose, an oxidative microenvironment was set up to simulate OA joint conditions in the elderly population. Hydrogen peroxide is a widely described model for oxidative stress in chondrocytes [48,49] and osteoblasts, and the consequent oxidative stress status inhibits the differentiation of osteoblast precursors [50,51,52]. Moreover, reduced amounts of antioxidants in aged people, mostly affected by OA, have been reported [53]. The results obtained using this model, based on the exposure of the osteoblasts seeded in the different scaffolds to oxidative stress, confirmed that cell viability and gene expression are affected by H_2_O_2_. The overall picture that emerges from a preliminary observation of cell/scaffold morphology highlights a double alteration in terms of both scaffolds and cells, confirming the aggressiveness of this in vitro condition [29]. All the biomaterials showed greater laxity, less compactness, and lower organization after the hydrogen peroxide treatment, possibly influencing the cell behavior in different ways. In particular, the Sr-ACP scaffold displayed the worst performance in terms of osteogenic potential. A possible reason could be the heterogeneity of the native scaffold structure together with the large dimensions of the Sr-ACP granules, which were bigger than the osteoblasts themselves. Furthermore, when this unstained scaffold was observed, a different presence and morphology of the Sr-ACP granules in the H_2_O_2_-treated condition were observed compared with the untreated one. It has been shown that H_2_O_2_ can react with mineralized tissues as well as with organic matter like cells or maybe even collagen [54,55]. Therefore, it could be speculated that H_2_O_2_ affects scaffold composition, being that the scaffold’s bony layer is composed of a mineral component of magnesium-substituted calcium hydroxyapatite with large Sr-ACP granules. It cannot be excluded that the treatment with hydrogen peroxide, even though mild, was responsible for the observed changes in the granules’ morphology, influencing cell behavior. On the contrary, the observation of the unstained OC+BMP-2 scaffold suggested a better ability to react to oxidative stimuli, preventing the detrimental consequences caused by H_2_O_2_ exposure. In agreement with this observation, the TEM analysis showed, in this scaffold, the presence of collagen fibrils, the most abundant organic component of the bone extracellular matrix, despite the treatment with H_2_O_2_. The small vesicles observed with the TEM in the cells on the OC and OC+BMP-2 scaffolds could also be an important sign of mineralization [56]. It is interesting to note that despite reduced cell viability in an aggressive microenvironment, there was an increased expression of the genes involved in matrix deposition (ALPL, COL1A1, SPARC, and BGLAP), though without reaching a statistically significant difference. By IHC, it was also possible to observe the cells’ effort to synthesize and secrete Type I Collagen. The higher expression of ALPL and COL1A1 produced by the cells grown on the OC+Sr-ACP was not accompanied by better collagen secretion, unlike what was observed on the other biomaterials. This could be attributable to the observed alteration of the organelles (e.g., the endoplasmic reticulum) and to the cell swelling phenomenon. Conversely, the presence of H_2_O_2_ allowed increases in COX2 and iNOS3, similarly to the effect elicited on cells alone. Overall, all the modified scaffolds seemed to exert a positive influence on osteoblasts in terms of typical genes’ expressions, but they did not influence the expressions of both genes related to inflammation. However, this should be considered as a limitation of the study. Maybe a deeper investigation of the oxidative stress cell status following the treatment with H_2_O_2_ would have given more information about the impacts of the different biomaterials on these conditions. Moreover, this study had a lack of quantification of the immunohistochemical collagen staining intensity, which would have provided a comprehensive understanding of the results. 

In conclusion, the BMP-2-augmented material seems to be the most promising solution in terms of osteoblast morphology, scaffold colonization, and activity. The addition of BMP-2 to the biomaterial seems to be able to improve osteoblast behavior in aggressive in vitro environments but also in normal conditions. Thus, this strategy clearly also represented an improvement of the bone-like layer of the OC scaffold in more challenging conditions, supporting further preclinical studies to optimize it for translation into clinical practice.

## 4. Materials and Methods

### 4.1. Scaffolds

The bone-like layer of a bilayered scaffold composed of 60% collagen and 40% collagen–magnesium–hydroxyapatite (Col/Col-Mg-HAp) (Maioregen, Fin-Ceramica, Faenza, Italy), which mimics the composition of subchondral bone, was used in this study. The BMP-2-modified OC scaffold (OC+BMP-2) was obtained through the adsorption of 4 µg of BMP-2 to the scaffold surface for 30 min at 37 °C. The Sr-ACP-modified scaffold (OC+Sr-ACP) was obtained by adding Sr-ACP granules to the scaffold’s Col/Col-Mg-HAp layer (30 *w*/*w*%). Sr-ACP granules were prepared by modifying a previously described synthesis method [57,58] by adding a Sr ion source (water-soluble Sr salt), and then the Sr-ACP was dried and crushed into granules of 100–150 μm in size.

### 4.2. Cell Cultures

Normal Human Osteoblasts (NHOsts), purchased from Lonza (Morrisville, NC, USA), were cultured in an appropriate Osteoblast Growth Medium (OGM, Lonza, Walkersville, MD, USA) containing 0.1% ascorbic acid and 0.1% gentamicin and supplemented with 10% fetal bovine serum (FBS). The cultures were expanded at 37 °C in a 5% CO_2_/95% air-controlled atmosphere.

### 4.3. Cell Seeding and H_2_O_2_ Treatment Induction

Cells (passage 6) were seeded on OC, OC+BMP-2, and OC+Sr-ACP scaffolds. Before cell seeding, the scaffolds, placed in a 24-well plate, were preconditioned with 50 µL of OGM for 1 h; then, 2 × 10^5^ NHOsts were seeded dropwise (35 μL) onto each material and let adhere for 2 h. Afterward, 1 mL of OGM was added to each sample.

In parallel, the cell/scaffold constructs were treated with H_2_O_2_ following a protocol of exposure defined in previous analyses (Appendix A). In detail, after 24 h from cell seeding, the medium was changed with fresh OGM containing 400 µM of H_2_O_2_ and left for 24 h. After additional 48 h, the untreated and treated samples were analyzed as described below. All experiments were performed three times.

### 4.4. Hematoxilin and Eosin (H&E) Staining

To evaluate cell morphology and scaffold colonization, the cell/scaffold constructs were rinsed with PBS, fixed in 4% paraformaldehyde (PFA), dehydrated in a graded series of alcohol (xylene 70%, 90%, 100%), embedded in paraffin, and finally cut into 5 μm thick sections with a rotary microtome (Microm HM355S, Thermo Fisher Scientific, Waltham, MA, USA). The slides were then deparaffined in xylene and graded concentrations of ethanol (100%, 95%, 85%, 75%), and rehydrated in distilled water, thenimmersed in hematoxylin solution for 10 min, distilled water, and finally, 1% Eosin solution for 1 min.

Unstained and stained sections were observed under an optical microscope (BX51; Olympus Italia, Milano, Italy) connected to an XC50 Olympus digital camera (Olympus Optical) and an image analyzer system (CellSens Dimension, Life Sciences Imaging Software, Olympus Italia). In parallel, the scaffolds treated as the cells were observed in order to analyze the possible effects of H_2_O_2_ on the scaffolds’ morphologies (Appendix A).

### 4.5. Immunohistochemistry (IHC)

To investigate the production of organic matrix by cells grown on scaffolds, an immunohistochemical staining for Type I Collagen (COL1A1) (MILLIPORE, Temecula, CA, USA) was performed. After endogenous peroxidase blocking and antigen unmasking, sections were incubated with primary mouse monoclonal anti-human antibodies overnight at 4 °C. Secondary antibodies (Vectastain Universal Quick kit; Vector Laboratories, Burlingame, CA, USA) and a development kit (Vector Nova Red; Vector Laboratories) were applied following the manufacturer’s instructions. Representative images were captured with an optical microscope (BX51, Olympus Italia) connected to an XC50 Olympus digital camera (Olympus Italia) and to an image analyzer system (CellSens Dimension, Life Sciences Imaging Software, Olympus Italia).

### 4.6. Transmission Electron Microscopy (TEM) Analysis

The cell/scaffold constructs and the scaffolds alone were subjected to an ultrastructural analysis. To this aim, they were fixed with 2.5% glutaraldehyde in 0.1 M of cacodylate buffer for 1 h at RT and 3 h at 4 °C. Afterward, the samples were washed with 0.1 M of cacodylate buffer, post-fixed with osmium tetroxide for 2 h, dehydrated in graded concentrations of ethanol and propylene oxide, and, finally, embedded in EPON 812. Ultrathin sections (80 nm) were stained with uranyl acetate and lead citrate and observed with a Jeol Jem-1011 electron microscope at 100 kV. Images were captured using an Olympus digital camera and iTEM Software version 5.1.

### 4.7. Alamar Blue Assay

To quantify the viability, or metabolic activity, of the cells grown on the scaffolds in normal and stress conditions, an Alamar blue assay was performed (Thermo Fisher Scientific). Briefly, dye was added to the cultures (1:10 *v*/*v*) and incubated for 4 h at 37 °C. This assay is based on the presence of an oxidation reduction (REDOX) indicator that changes color in response to chemical reductions induced by the mitochondria of active cells. The resulting fluorescence was read at 530ex–590em nm wavelengths with a Micro Plate reader (VICTOR X2030, Perkin Elmer, Milano, Italy) and expressed as relative fluorescence units (RFUs). A culture medium with a reagent and without cells was used as a control for background fluorescence. In addition, since the chemical nature of the material tended to retain part of the reagent, an additional 3 h incubation with the growth medium only was performed to allow the release of the retained reagent, which was then quantified.

### 4.8. Gene Expression

The expression levels of COL1A1, Alkaline Phosphatase (ALPL), Osteocalcin (BGLAP), and Osteonectin (SPARC) as osteogenic markers and of Cyclooxygenase 2 (COX2) and Nitric Oxide Synthase Type 2 (iNOS2) as inflammation markers were assessed with Real-Time semiquantitative PCR (qPCR) 48 h after H_2_O_2_ stimulation.

Total RNA was extracted from NHOsts grown on scaffolds following the TRIzol/chloroform method until the harvesting of the aqueous phase. In more detail, 1 mL of TRIzol reagent (Ambion, Life Technologies, Carlsbad, CA, USA) was added to each sample and incubated for 5 min at room temperature. Chloroform was then added in a volume equal to 1/5 of the TRIzol and carefully mixed. After centrifugation at 12,000 rpm at 4 °C, the aqueous phase was collected, added to 75% cold ethanol and then processed using the commercial Rneasy Mini Kit (Purelink™ RNA miniKit; Ambion, Life Technologies, Carlsbad, CA, USA). The obtained RNA was then quantified with a NanoDrop spectrophotometer (Thermo Fisher Scientific) and reverse transcribed using the Superscript Vilo cDNA synthesis kit (Life Technologies) following the manufacturer’s instructions. Ten ng of cDNA were tested in duplicate for each sample. Gene expression was evaluated with semiquantitative Real-Time PCR analysis using a SYBR green PCR kit (Qiagen GmbH, Hilden, Germany) in a Light Cycler 2.0 Instrument (Roche Diagnostics, GmbH, Manheim, Germany). The protocol included a denaturation cycle at 95 °C for 15 min, 25 to 40 cycles of amplification (95 °C for 15 s, an appropriate annealing temperature for each target for 20 s, and 72 °C for 20 s), and a melting curve analysis to check for amplicon specificity.

The primers (Qiagen) used to evaluate gene expression are detailed in Table 1. The mean threshold cycle was determined for each sample and used to calculate relative expression using the Livak method (2^−ΔΔCt^), with GAPDH as the reference gene [59]. To compare the groups, untreated cells on the OC scaffold were used as calibrators.

### 4.9. Statistical Analysis

Data were analyzed using R software v 4.2.2 (31 October 2022) [60] and the ‘emmeans’ package, version 1.8.3 (8 March 2023) [61]. After verification of normal distribution (Shapiro–Wilk test), two-way ANOVA was used to compare data among groups, followed by a pairwise multiple comparison test considering ‘scaffold’ (OC, OC+BMP-2, and OC+Sr-ACP) and ‘H_2_O_2_ treatment’ (yes, no) as fixed effects. *p*-values were adjusted according to Sidak’s method. Data were reported as means ± SD at a significance level of *p* < 0.05.

## Figures and Tables

**Figure 1 ijms-24-14764-f001:**
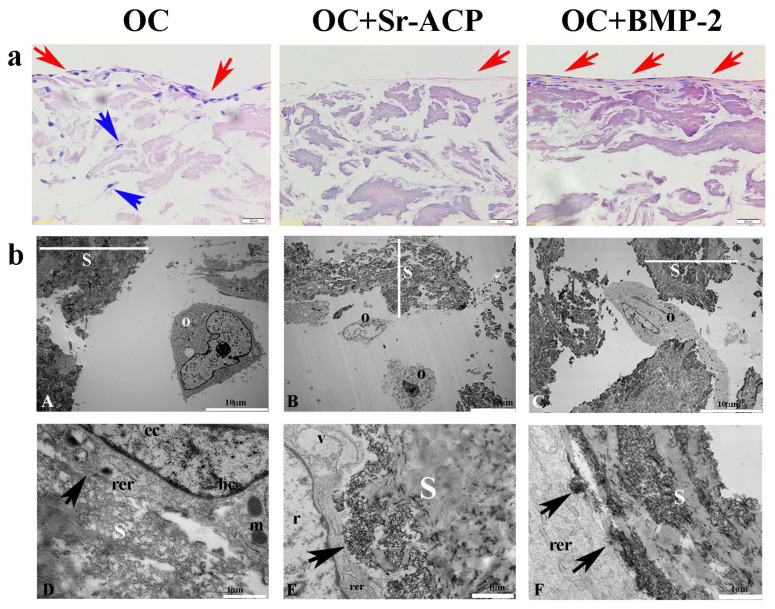
NHOst morphology in the different scaffolds. (**a**) H&E staining. Representative images. Magnification: 20×; scale bar: 20 µm. (**b**) Representative TEM images of NHOst cells/scaffold interactions. rer indicates a rough endoplasmic reticulum, m indicates mitochondria, hc indicates a heterochromatin compartment, ec indicates a euchromatin compartment, r indicates rarefied chromatin, v indicates vacuoles, black arrows indicate focal contacts, s indicates a scaffold, O indicates osteoblasts, and red and blue arrows indicate the presence of cells on the scaffold surface and inner part, respectively. (**A**,**C**) Scale bar: 10 µm; (**B**,**D**–**F**) scale bar: 1 µm.

**Figure 2 ijms-24-14764-f002:**
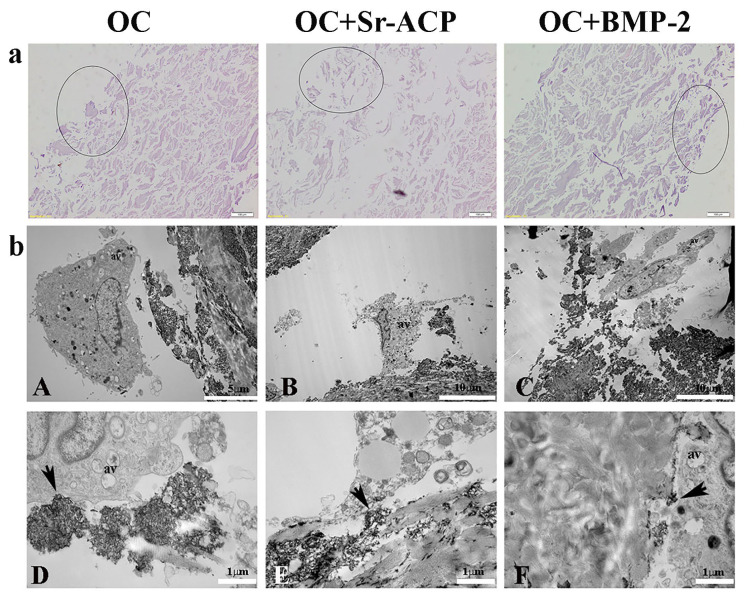
NHOst morphologies of the different scaffolds after H_2_O_2_ treatment. (**a**) H&E staining. Representative images. Circles indicate the areas colonized by osteoblasts. Magnification: 20×; scale bar: 20 µm. (**b**) Representative TEM images of NHOst cells/biomaterial interactions after H_2_O_2_ exposure. Black arrows indicate focal contacts, av indicates autophagic vacuoles. (**A**) Scale bar: 5 µm; (**B**,**C**) scale bar: 10 µm; (**D**–**F**) scale bar: 1 µm.

**Figure 3 ijms-24-14764-f003:**
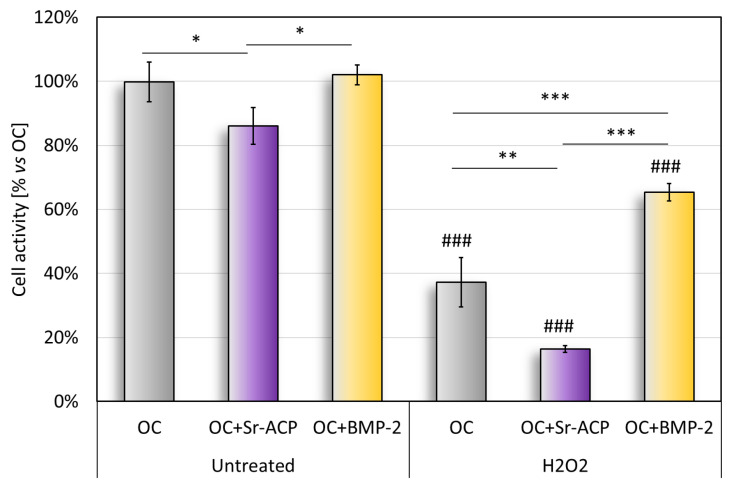
Alamar blue assay. Cell metabolic activity after 96 h of cell culture on scaffolds with and without H_2_O_2_ treatment, reported as a percentage, considered as 100%, with respect to OC. Mean ± SD, *n* = 3. * *p* < 0.05, ** *p* < 0.005, *** *p* < 0.0005), ### *p* < 0.0005 vs. untreated.

**Figure 4 ijms-24-14764-f004:**
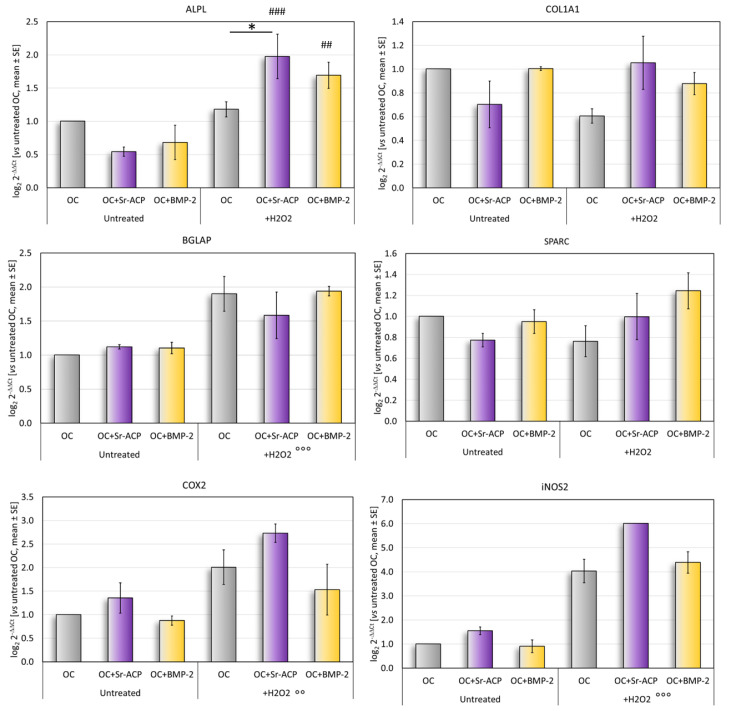
Gene expression analysis. Relative gene expressions of ALPL, COL1A1, SPARC, BGLAP, COX2, and iNOS2 after 72 h of culture on scaffolds, with and without H_2_O_2_, reported as fold changes with respect to cells grown on OC scaffold, considered to be 1. Mean ± SD. * *p* < 0.05. ## *p* < 0.005 vs. untreated, ### *p* < 0.0005 vs. untreated; for all materials: °° *p* < 0.005 vs. untreated, °°° *p* < 0.0005 vs. untreated.

**Figure 5 ijms-24-14764-f005:**
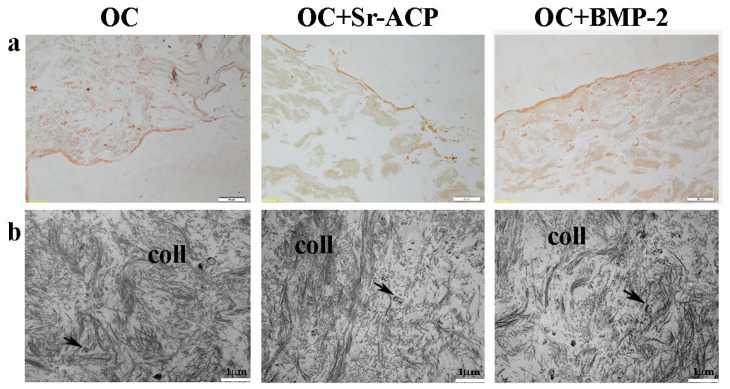
Representative IHC images for Type I Collagen synthesis by NHOst cells seeded on OC, OC+BMP-2, and OC+Sr-ACP scaffolds. (**a**) IHC staining. Magnification: 10×; scale bar: 20 µm. (**b**) TEM images of matrix deposition. coll indicates collagen fibrils; black arrows indicate small vesicles. Scale bar: 1 µm.

**Figure 6 ijms-24-14764-f006:**
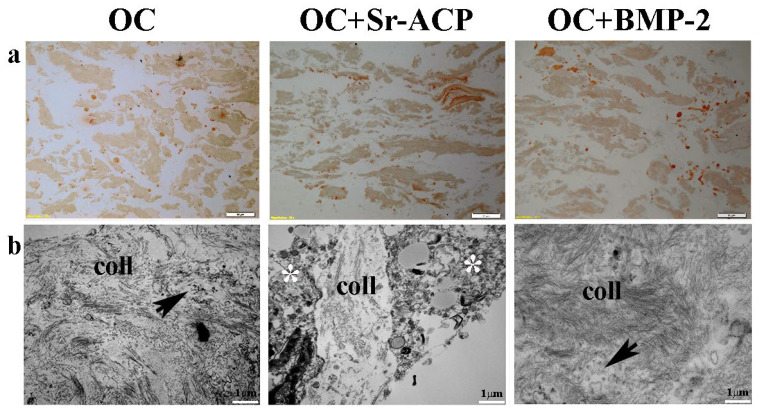
Representative IHC images for Type I Collagen synthesis by NHOst cells seeded on OC, OC+BMP-2, and OC+Sr-ACP scaffolds after H_2_O_2_ exposure. (**a**) IHC staining. Magnification: 10×; scale bar: 20 µm. (**b**) TEM images of matrix deposition. coll indicates collagen fibrils, black arrows indicate small vesicles, and asterisks indicate cytoplasmic swelling. Scale bar: 1 µm.

**Table 1 ijms-24-14764-t001:** Primers and protocol used for the Real-Time PCR assay.

Gene	Forward and Reverse Primers	Amplicon Length (bp)	Annealing Temperature (°C)
GAPDH	5′-TGGTATCGTGGAAGGACTCA-3′ 5′-GCAGGGATGATGTTCTGGA-3′	123	56
COL1A1	QuantiTect Primer Assay (Qiagen) Hs_COL1A1_1_SG	118	55
ALPL	QuantiTect Primer Assay (Qiagen) Hs_ALPL_1_SG	110	55
BGLAP	QuantiTect Primer Assay (Qiagen) Hs_BGLAP_1_SG	90	55
SPARC	QuantiTect Primer Assay (Qiagen) Hs_SPARC_1_SG	60	55
COX2	QuantiTect Primer Assay (Qiagen) Hs_PTGS2_1_SG	68	55
iNOS2	QuantiTect Primer Assay (Qiagen) Hs_NOS2_1_SG	92	55

## Data Availability

Not applicable.

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
