# Peer review of "Human Osteoblasts’ Response to Biomaterials for Subchondral Bone Regeneration in Standard and Aggressive Environments"

_ijms, 2023, doi:10.3390/ijms241914764_

Round 1

Reviewer 1 Report

Stefania etc. have presented an evaluation of a bilayered scaffold that mirrors the composition of the subchondral bone, predominantly composed of collagen and collagen-magnesium-hydroxyapatite. Using Normal Human Osteoblasts, the study assessed the scaffold's efficacy under various conditions, including oxidative stress scenarios. Notably, the scaffold demonstrated support for osteoblast growth and differentiation, indicating its potential utility in tissue engineering and relevant clinical settings. While the paper presents an intriguing contribution to bone tissue engineering, I have several suggestions for improvement:

1. A concise discussion on the potential implications of such scaffolds in aged bone environments would be insightful. It would be interesting to consider how alterations in bone marrow adiposity and osteogenesis due to aging could impact the efficacy of the scaffold.

2. To offer better context, particularly for readers with a background in aging research, it might be beneficial if the results were juxtaposed against how these metrics typically shift with age. For example, is it possible that the incorporation of BMP-2 might counteract age-related declines in osteogenesis?

3. Given that aging is often linked with elevated adipocyte accumulation in the bone marrow, which subsequently reduces osteogenesis, does the presented scaffold exhibit any resilience against these alterations? More specifically, does the scaffold's emulation of subchondral bone create a favorable milieu for osteoblast functionality, even amidst age-related transformations?

4. In Figure 1A, the displayed image lacks clarity, and the magnification seems insufficient. I would recommend providing images with higher magnification that spotlight osteoclasts.

For Figure 1, a histological analysis of the osteoblast would be valuable, preferably showcasing Ob.N/BS.

In a few areas, the language becomes slightly redundant. Streamlining the language will make the paper more concise and improve clarity. Overall, the quality of English is good, but could benefit from editing to improve clarity, correct minor errors, and enhance readability.

Author Response

RESPONSE TO REVIEWER 1 COMMENTS

 We thank you very much for reviewing our manuscript and for the precious suggestions, giving us the opportunity to improve our manuscript. Please, find here the point-by-point response. We modified the manuscript according to your requests and provided new high-resolution figures. 

We hope that the manuscript is now improved, more interesting for the readers, and suitable for publication, but please let us know if there is anything else we should do to finalize this manuscript.

We look forward to hearing from you.

  1. A concise discussion on the potential implications of such scaffolds in aged bone environments would be insightful. It would be interesting to consider how alterations in bone marrow adiposity and osteogenesis due to aging could impact the efficacy of the scaffold.

Very interesting questions, related to the importance of the bone marrow as stem cell niche for and source of osteoblast precursors, which have a key role in the osteochondral regeneration process. This is of particular interest for us, as our model was aimed at exploring the osteoblast behaviour also in a challenging context, like aged environment.

The imbalance of the medullary niche during aging process has become a very hot biologic topic in the recent years, as well as the features of bone marrow mesenchymal cells (BMSCs) may change with patients’ age, as they decrease in number, clonogenic ability, and differentiative potential [PMIDs: 29751774; 35485439]. At the same time, however, some Authors argue that mature osteoblasts in stressed conditions tend to compensate the decreased bone formation with hyperactivity [26220824], BMSCs maintain the same clonogenic and differentiating ability in young, old, and osteoporotic patients [11393789], or adipose derived mesenchymal cells (ADSCs) are able to maintain their differentiating ability in osteoporotic condition [24687841]. We added these considerations in the Discussion section (lines 245-251)

We are currently performing another study focused on the investigation of the behaviour of osteoblast precursors, MSCs, on the same scaffolds, and we will take into account this important issue to better explore the scaffolds’ potential.

  1. To offer better context, particularly for readers with a background in aging research, it might be beneficial if the results were juxtaposed against how these metrics typically shift with age. For example, is it possible that the incorporation of BMP-2 might counteract age-related declines in osteogenesis?

Regarding BMP-2, there is substantial agreement on the positive osteogenic role of this factor [PMIDs: 19892583; 32730937], sometimes also in an aged microenvironment [PMID: 11453116]. In the study of Matsumoto et al, in particular, the biomaterial was associated with recombinant BMP-2.

The results of other Authors, which described an imbalanced expression of BMP2/4 signalling in an aged bone environment, thus favoring the adipogenic differentiation, confirmed the importance of the BMP-2 environment enrichment. In this light, we think that the in vitro good results obtained in the H2O2-treated samples with the BMP-2 scaffold could suggest that this approach could favour the in vivo regeneration on subchondral bone in patients with age-related declines in osteogenesis.

We added these considerations in the Discussion section (lines 261-264).

  1. Given that aging is often linked with elevated adipocyte accumulation in the bone marrow, which subsequently reduces osteogenesis, does the presented scaffold exhibit any resilience against these alterations? More specifically, does the scaffold's emulation of subchondral bone create a favorable milieu for osteoblast functionality, even amidst age-related transformations?

Actually, we do not have any in vivo study on aged subjects with the modified scaffolds. However, the clinical outcome of patients treated with the unmodified scaffold was not influenced by age (PMIDs: 32909451; 25457331). We added this consideration also in the Discussion section (lines 272-275).

  1. In Figure 1A, the displayed image lacks clarity, and the magnification seems insufficient. I would recommend providing images with higher magnification that spotlight osteoclasts. For Figure 1, a histological analysis of the osteoblast would be valuable, preferably showcasing Ob.N/BS.

We agree with the Reviewer. At this regard, we improved the quality of Figure 1, as well as of Figures 2, 5, 6, and Supplementary Figure 1. We chose the 20X magnification to get a broader view of the scaffold colonization (osteoblasts grown of the surface and inner part of the differently functionalized scaffolds). Accordingly, we assessed the H&E staining to observe cell distribution as well as morphology- also analysed at an ultra-high magnification by TEM. Unfortunately, it was not possible to count the number of osteoblasts as they grow on a dense monolayer. For this reason, we then assessed an indirect assay (Alamar Blue assay) to obtain quantitative information on cell number.

  1. In a few areas, the language becomes slightly redundant. Streamlining the language will make the paper more concise and improve clarity. Overall, the quality of English is good, but could benefit from editing to improve clarity, correct minor errors, and enhance readability.

The English language has been revised and corrected by a native speaker.

Reviewer 2 Report

The authors modified the collagen/collagen-magnesium-hydroxyapatite (Col/Col-Mg-HAp) scaffold by incorporating BMP-2 and Sr, and subsequently assessed the scaffolds' osteogenic potential in vitro. In my opinion, this manuscript has certain weaknesses in terms of both experimental design and figure presentation. As it stands, it may not be suitable for publication in its current format. The specific comments are outlined below:

1. The manuscript emphasizes the osteochondral (OC) unit in the title and introduction, but the study primarily focuses on detecting the osteoblastic differentiation of human osteoblasts. It is essential to align the study's objectives with the emphasized OC unit and its potential.

2.While the H2O2 treatment aims to establish oxidative stress, there is a lack of confirmed markers for oxidative stress status. Additionally, the impact of different biomaterials on this status is not explored.

3.The resolution of the figures, including the H&E staining images, is notably low. Enhancing the figures' quality will aid in the accurate interpretation of results.

4.Figures 5 and 6 lack an analysis of immunohistochemical staining intensity. Incorporating this analysis will provide a comprehensive understanding of the experimental outcomes.

5.It is advisable to include the release profiles of ions and BMP-2 from the biomaterials. This data is crucial for understanding the scaffolds' behavior and potential applications.

6.Investigating the mechanical characteristics of the scaffolds will further enhance the study's comprehensiveness. Mechanical properties play a significant role in evaluating the suitability of biomaterials.

7.Clarification is needed regarding the abbreviation "OC." In the abstract (line 23), "OC" is intended to abbreviate "collagen/collagen-magnesium-hydroxyapatite (Col/Col-Mg-HAp) scaffold." However, throughout the manuscript, "OC" seems to have a different connotation. In the introduction (line 40), "OC" appears to stand for "osteochondral." It's essential to clarify these abbreviations consistently throughout the manuscript.

Moderate editing of English language required

Author Response

RESPONSE TO REVIEWER 2 COMMENTS

We thank you very much for reviewing our manuscript and for the precious suggestions, giving us the opportunity to improve our manuscript. Please, find here the point-by-point response. We modified the manuscript according to your requests and provided new high-resolution figures.

We hope that the manuscript is now improved, more interesting for the readers, and suitable for publication, but please let us know if there is anything else we should do to finalize this manuscript.

We look forward to hearing from you.

The authors modified the collagen/collagen-magnesium-hydroxyapatite (Col/Col-Mg-HAp) scaffold by incorporating BMP-2 and Sr, and subsequently assessed the scaffolds' osteogenic potential in vitro. In my opinion, this manuscript has certain weaknesses in terms of both experimental design and figure presentation. As it stands, it may not be suitable for publication in its current format. The specific comments are outlined below:

  1. The manuscript emphasizes the osteochondral (OC) unit in the title and introduction, but the study primarily focuses on detecting the osteoblastic differentiation of human osteoblasts. It is essential to align the study's objectives with the emphasized OC unit and its potential.

 We agree with the Reviewer: the study aim with respect to the osteochondral unit function was not enough clear.

The scaffold used in this study is a bilayered, with one cartilage-like layer, and one bone-like layer which mimics the composition of the subchondral bone. Its clinical application regards the regeneration of the entire osteochondral unit, showing very good clinical results in terms of new cartilage formation but suboptimal in terms of osteochondral bone regeneration. Therefore, the critical point is the incomplete bone formation, and the improvement of osteoblast differentiation is the goal of the two different scaffold modifications of the bone-like layer. For this reason, we limited the analysis to the osteoblast growth on the bone-like layer and to their differentiation. We clarified our aim in the Abstract, Introduction, and Materials and Methods sections (lines 25-26, 28-30, 60-63, 87, 236-237, 358).

  1. While the H2O2 treatment aims to establish oxidative stress, there is a lack of confirmed markers for oxidative stress status. Additionally, the impact of different biomaterials on this status is not explored.

We thank the Reviewer for this observation, coherent with the topic of the study. The main aim of the scaffold modifications was to stimulate and improve the regeneration of subchondral bone in an aged microenvironment. The aged environment was simulated by the addition of H2O2, as reported in literature.

We focused on the main aspects related to osteoblasts in relationship to the new materials and used the H2O2 as a model to stress the cells. Scaffold colonization, cell viability and morphology, cell-material adhesion, gene expression of the main osteogenic markers and matrix synthesis are the readout parameters for effect.

An analysis of the oxidative stress status was indeed not performed, however, in order to preliminarily assess this aspect the expression of COX-2 and iNOS2 was evaluated. We added a sentence on this regard in the Discussion section (lines 341-344).

  1. The resolution of the figures, including the H&E staining images, is notably low. Enhancing the figures' quality will aid in the accurate interpretation of results.

We agree with the Reviewer. The resolution of Figures 1, 2, 5, 6, and Supplementary Figure 1 has been improved.

  1. Figures 5 and 6 lack an analysis of immunohistochemical staining intensity. Incorporating this analysis will provide a comprehensive understanding of the experimental outcomes.

We thank the Reviewer for this comment. We have performed an image analysis based on the measurement of the stained areas with respect to the region of interest (ROI) and here is the result (Type I coll intensity/ROI):

Untreated

+H2O2

OC

3.03 %                                      

1.23 %

OC+Sr-ACP

1.59 %

1.52 %

OC+BMP-2

4.73 %                        

1.55 %

However, we do not feel comfortable to incorporate this data into the paper since to be able to quantify intensity of immunohistochemical stainings many control and validation experiments should be performed. Since we acknowledge that this information could have helped in the understanding of the results, s we added this point in the Discussion section as a limitation (lines 344-346)

  1. It is advisable to include the release profiles of ions and BMP-2 from the biomaterials. This data is crucial for understanding the scaffolds' behavior and potential applications.

Thank you for the very pertinent question. Unfortunately, we do not have performed experiments on the release of ions from this scaffold yet. However, the release profile of BMP-2 from the same scaffold has been already investigated in a previous study (PMID: 36826910). In particular, the in vitro release kinetics of BMP-2 from both the bone-like layer (the same used in our study) and collagen-like layer of the osteochondral scaffold over 14 days has been assessed. The results showed that BMP-2 was largely retained in both layers, especially in the bony layer, avoiding a too fast release. Therefore, a sustained in vivo release should be expected, avoiding potential side effects of BMP-2 while helping at the same time the regeneration of the osteochondral bone tissue. Unfortunately, in vivo other factors might influence the release kinetics (i.e. the scaffold degradation time) making correlations between in vitro and in vivo unreliable.

  1. Investigating the mechanical characteristics of the scaffolds will further enhance the study's comprehensiveness. Mechanical properties play a significant role in evaluating the suitability of biomaterials.

The osteochondral tissue is complex, entailing both articular (hyaline) cartilage and subchondral bone, in which the mechanical properties gradually change, according to the mineral content. This clinically used scaffold is a three-layered one, made by the combination of distinct but integrated layers with different composition and mechanical properties mimicking the native cartilage and bone regions of the osteochondral unit (we tested the bone-layer). This scaffold is softer and less viscoelastic than the natural tissue. Cells proliferate, differentiate, and synthesize matrix according to scaffold layer composition. The progressive resorption of the scaffold and the simultaneous cell mediated remodelling favour the complete regeneration of the tissue. In clinical application, this scaffold is implanted in patients with osteochondral defects, and it is not subjected to mechanical loading for about 6 weeks and a mechanical support is also applied. An extensive scaffold characterization from this point of view has been performed previously (PMIDs: 14566805, 18538387, 31029997).

  1. Clarification is needed regarding the abbreviation "OC." In the abstract (line 23), "OC" is intended to abbreviate "collagen/collagen-magnesium-hydroxyapatite (Col/Col-Mg-HAp) scaffold." However, throughout the manuscript, "OC" seems to have a different connotation. In the introduction (line 40), "OC" appears to stand for "osteochondral." It's essential to clarify these abbreviations consistently throughout the manuscript.

Thank you for the suggestion, we removed the abbreviation OC relative to “osteochondral” throughout the text. We are now using it only when referring to the scaffold.

  1. Moderate editing of English language required

The English language has been revised and corrected by a native speaker.

Round 2

Reviewer 1 Report

The author responded well to the previous questions; the language and content of the article have improved compared to before, meeting the publication requirements.

The English language of the article is fluent, precise, and reaches the publication standard.

Author Response

RESPONSE TO REVIEWER 1 COMMENTS

We appreciated very much the attention you paid to all the review process. Your insightful comments and suggestion have undoubtedly improved the quality of our manuscript. We hope that the Manuscript finally deserves to be published in IJMS.

Once again, we thank you for your invaluable support.

Comments and Suggestions for Authors

The author responded well to the previous questions; the language and content of the article have improved compared to before, meeting the publication requirements.

Comments on the Quality of English Language

The English language of the article is fluent, precise, and reaches the publication standard.

We thank the Reviewer for the positive feedback.
